# Marine-Derived Secondary Metabolites as Promising Epigenetic Bio-Compounds for Anticancer Therapy

**DOI:** 10.3390/md19010015

**Published:** 2020-12-31

**Authors:** Mariarosaria Conte, Elisabetta Fontana, Angela Nebbioso, Lucia Altucci

**Affiliations:** Department of Precision Medicine, University of Campania ‘Luigi Vanvitelli’, Via L. De Crecchio 7, 80138 Naples, Italy; elisabetta.fontana@unicampania.it (E.F.); angela.nebbioso@unicampania.it (A.N.)

**Keywords:** secondary metabolites, epigenome, epigenetic signaling, bioactive compounds, cancer therapy, marine species, environment

## Abstract

Sessile organisms such as seaweeds, corals, and sponges continuously adapt to both abiotic and biotic components of the ecosystem. This extremely complex and dynamic process often results in different forms of competition to ensure the maintenance of an ecological niche suitable for survival. A high percentage of marine species have evolved to synthesize biologically active molecules, termed secondary metabolites, as a defense mechanism against the external environment. These natural products and their derivatives may play modulatory roles in the epigenome and in disease-associated epigenetic machinery. Epigenetic modifications also represent a form of adaptation to the environment and confer a competitive advantage to marine species by mediating the production of complex chemical molecules with potential clinical implications. Bioactive compounds are able to interfere with epigenetic targets by regulating key transcriptional factors involved in the hallmarks of cancer through orchestrated molecular mechanisms, which also establish signaling interactions of the tumor microenvironment crucial to cancer phenotypes. In this review, we discuss the current understanding of secondary metabolites derived from marine organisms and their synthetic derivatives as epigenetic modulators, highlighting advantages and limitations, as well as potential strategies to improve cancer treatment.

## 1. Introduction

Marine habitats are an extraordinary source of new and structurally complex bioactive metabolites naturally produced by different organisms and characterized by unique functions with marked biological activities. These features can be attributed to extreme environmental conditions such as lack of light, high pressure, ionic concentration, pH and temperature changes, scarcity of nutrients, and restricted living spaces [1]. The high concentration of coexisting organisms in a limited area also makes them very competitive and complex, resulting in the development of adaptations and behaviors aimed at safeguarding the species, such as the adoption of chemical strategies exploiting the wealth of bioactive molecules produced by the secondary metabolism [2,3]. Marine-derived metabolites originate from different signal transduction pathways activated as a consequence of epigenome changes in the organisms that produce them. Phenotypic/genotypic alterations of marine organisms are characterized by an intricate network of interactions that influence each other. Such interplay is further complicated by epigenetic modifications, which can trigger adaptive biochemical processes in the species [4]. The marine environment (characterized by biotic and abiotic factors), in turn, plays an essentially selective role in intrinsically changing organisms, exerting an inductive function on epigenetic, genetic, and phenotypic changes with transgenerational effects on the species [5,6] (Figure 1). Since the reprogramming of epigenetic states can be induced by environmental exposures in the marine habitat, secondary metabolites produced by a large number of organisms might represent good candidates as novel natural molecules with potential pharmacological activity for cancer treatment [7]. Cutting-edge chromatographic isolation and purification techniques, pharmacological screening methods, and numerous spectroscopic approaches for structural investigation such as mass spectroscopy and nuclear magnetic resonance (NMR) were used to isolate and characterize several new marine-derived compounds [8,9,10], some of which have potential anticancer activities. The chemical composition of isolated molecules has a major impact on both the epigenome of the organism [11] and any potential epigenetic effects produced, reflecting a complex and interconnected machinery of information exchange. In this review, we describe the therapeutic potential of marine-derived secondary metabolites and their synthetic derivatives in cancer, focusing on their importance as epigenetic modulators generating posttranscriptional, inductive (produced by the metabolism of the organism), and induced (produced by alterations in marine environment) modifications. We also discuss the challenges involved in discovering new natural and synthetic marine bio-compounds with anticancer activity in light of the enormous variability that characterizes the organisms themselves and the environment that surrounds them. This review also highlights sustainable use of marine resources as producers of high yields of value added bio-molecules for pharmaceutical field towards a more sustainability of economic growth in terms of development, research and transmissibility of marine technology in terms of development, research, and transmissibility of marine technology.

## 2. Anticancer Activities of Marine-Derived Secondary Metabolites with Inductive and Induced Epigenetic Modifications

Epigenetics belongs to a branch of genetics based on the concomitance of complex biomolecular mechanisms, which coordinate genetic information in the nucleus, culminating in control of gene expression [12], which in turn propagates in subsequent generations. All this information is further conditioned by perturbations from the external environment. Epigenetic alterations in gene activity are mitotically stable in the absence of changes in DNA sequence [13]. Generally, mechanisms of environmental perception act through alterations in chemical tags that normally exist in the genome. In cancer, “epigenetic markers” [14] serve as a sort of barcode of DNA function, indicating whether genes are active or silent. The alteration and reprogramming of epi-signals can lead to changes in gene expression and also directly influence transcriptional regulator function, with downstream effects on the way cells and tissues work. In addition to genetic alterations, a hallmark of various types of cancer, epigenetic dysregulations affecting DNA methylation, histone modifications, and microRNAs introduce another layer of complexity, contributing to tumor progression and changes in the phenotypic state. These epi-alterations are further regulated by so-called chromatin writers, readers, and erasers, which constitute specialized protein machinery able to modulate and reversibly influence the epigenome. DNA methyltransferases (DNMTs), histone acetyltransferases (HATs), histone methyltransferases (HMTs), and lysine/arginine methyltransferases (KMTs/RMTs) are all writers, due to their ability to add a modification on DNA and histones; readers, able to “read” and thus interpret covalent modifications include: bromodomains, specific for acetylation site recognition; chromodomains, recognized by methyl-readers; methyl-lysine readers such as ATRX-DNMT3-DNMT3L (ADD), ankyrin, bromo-adjacent homology, chromobarrel, WD40, and zinc finger CW domains, as well as double chromodomain, and tandem Tudor domain. In addition, plant homeodomains bind to methylated histone H3, while the PWWP domain can bind DNA and methylated histones. Erasers include TET proteins, which remove modifications from DNA and histones, as well as histone demethylases (HDMs), histone deacetylases (HDACs), protein phosphatases, and deubiquitinating enzymes, which remove methyl, acetyl, phosphate, and ubiquitin groups from histones and other proteins, respectively [15,16,17]. Since modern medical approaches are based on the personalization of human healthcare, bioactive molecules with epigenetic activity isolated from marine sources represent a valid alternative to conventional therapies for use in extensive preclinical assessments and the advanced phases of clinical studies. Furthermore, given the resistance developed by some pathogens to pharmacological treatments and the inefficacy of traditional chemotherapies, efforts are being made to identify more biologically active and effective molecules [18]. In marine ecosystems, sessile organisms are much more susceptible to changes in the external environment [19] and adopt complex survival strategies. Moreover, the set of biotic and abiotic components in these organisms are extremely predominant, determining the production of secondary metabolites with almost unique chemical-physical characteristics. A further level of complexity is added by the intricate relationship between secondary metabolites and epigenetic functions, which in turn contribute to the development of defense mechanisms by the species that are transmitted across generations [20]. Secondary metabolites can harbor several beneficial properties for human health such as antioxidant, antibacterial, antivirus, anticoagulant, antidiabetic, anti-inflammatory, antihypertensive, and antitumor activities [21]. Furthermore, their natural biological functions are strongly influenced by the surrounding environment, including conditions of climatic stress or attack by predators. Computational programs using knowledge-based algorithms or sequence-based prediction [22] have identified genes responsible for the production of these natural products, but only for some species. These genes are usually located in specific biosynthetic gene clusters (BGCs) in the genome [23] that contain the required enzymes responsible for synthesis of secondary metabolites and regulatory structures. Considering the enormous genetic and epigenetic variability among marine species, it is not always possible to predict their BGCs and therefore their association with the production of secondary metabolites. For example, under certain conditions chromatin remodeling factors can switch on or switch off specific genes continuously over time. Cutting-edge technologies such as those involved in triggering the activation of silent BGCs, which include changes in growth conditions (e.g., temperature and pH) or genetic engineering-based approaches [24,25] are emerging to better study the interaction between the production of metabolites and the genes that produce them. One of the goals of anticancer research is to extract and select biomolecules from these organisms in order to exploit their properties and generate synthetic analogues (Figure 2). To date, the U.S. Food and Drug Administration (FDA) has approved several marine-derived therapeutic compounds such as cytarabine, vidarabine, ziconotide, omega-3 acid ethyl esters, eribulin mesylate, brentuximab vedotin, and iota-carrageenan [26,27,28,29,30,31,32] (Figure 3) and further studies aimed at characterizing and developing new drugs are ongoing. The following section describes the epigenetic role of well-known marine-derived secondary metabolites, classified according to their biosynthetic pathways and subdivided into three major families: phenolic compounds, cyclic peptides, and alkaloids. Their mechanism of action as potential epigenetic bio-compounds for the treatment of different type of cancers is also discussed.

## 3. Sustainability and Health

Potential anticancer drugs derived from various marine species are not always present in the environment in sufficient quantities and do not maintain the same functional characteristics over time both in terms of chemical-physical structure and biological potential, thus limiting their characterization. These molecules, before being used as drugs, need to be submitted to rigorous scientific research and to quality control according to precise standards and procedures created ad hoc to ensure their best implementation. One of the aspects that influences their production is represented by the conditions of the marine environment, which has a fundamental impact on development, research and on new strategies applied to marine biotechnology. The sea must be considered not only an environment to be exploited, but also to be safeguarded, since its protection has crucial benefits for human health. Apart from human activities, climate change, the availability of nutrients, the attack of predators also strongly affecting the production of bioactive compounds, for which we are increasingly trying to enhance sustainability, well-being and health, both in environmental protection and in socio-economic terms.

## 4. (Poly)phenolic Compounds

(Poly)phenolic compounds are one of the main classes of marine-derived secondary metabolites. They can be found in different pelagic organisms and their production varies across genera as well as growing conditions, geographical location, and abiotic/biotic factors. These compounds can be distinguished by the presence of one (phenolic acids) or more (polyphenols) aromatic rings annexed to hydroxyl groups in their structures, which confer very strong antioxidant properties. Their bioactivity is also linked to other enzymatic inhibitory effects as well as to anticancer, antidiabetic, or anti-inflammatory actions, with beneficial results for human health [33,34,35]. In addition, their role as scavengers of singlet oxygen and free radicals and/or reducing and chelating agents is a very promising area for the study and treatment of cancer, as they display interesting epigenetic molecular mechanisms that modulate gene expression as well as DNA damage and repair. The following subsections focus on the most studied natural phenolic compounds and their derivatives in terms of their epigenetic role in cancer and their use in clinical trials (Table 1 and Appendix A).

### 4.1. Psammaplin A

Psammaplins belong to a group of bromotyrosine phenols, whose common ancestor is Psammaplin A (PsA), a natural phenolic product isolated for the first time from the *Psammaplin aplysilla* marine sponge and from an unknown sponge (probably *Thorectopsamma xana*) in 1987 [36]. This marine metabolite was the first natural product containing oxime and disulfide moieties to be isolated from a marine sponge [29,37], and is characterized by a disulfide bridge and a bromotyrosine ring occurring in nature in the form of monomers or dimers. PsA exhibits anticancer activities by modulating different human enzymes, which in turn regulate DNA replication, transcription, differentiation, apoptosis, proliferation, tumor invasion, and migration. PsA also inhibits topoisomerase II, aminopeptidase N, chitinases, farnesyl protein transferase, leucine aminopeptidase, and other enzymes [36,38,39,40,41]. PsA is reported to act as an antiproliferative agent in various human cancer cell lines, such as endometrial, breast, and triple negative metastatic breast cancer, as well as in in vivo models [15,42,43] by exerting potentially inhibitory effects on HDACs and DNMTs. PsA was also shown to sensitize human lung and glioblastoma cancer cells to radiation in vitro; PsA pretreatment in these cells increased the sub-G1 phase of the cell cycle, induced an increased expression of cleaved caspase-3, and led to a drastic depletion of DNMT1 and DNMT3A, suggesting inhibition of the DNA damage repair process elicited by the DNA damage marker γH2AX [44]. The mechanism of action underlying the HDAC inhibitory effect of PsA involves a change in the redox state of the disulfide bond. Replacement of the sulfur atom leads to the formation of a mercaptan, which in turn chelates the Zn^+^ ion present in the characteristic active site of the HDAC enzyme, modifying its conformational state and thus preventing its accessibility to the natural substrate [45]. This new conformational state determines an increase in acetylation levels of histone H3, a well-known epigenetic marker of chromatin structure and function, suggesting selectivity for HDACs.

### 4.2. Indole-Derived Psammaplin a Analogues

Because of the limits relating to the extraction and instability of PsA, several indole-derived analogues have been designed and many studies undertaken to improve the inhibitory effect of this promising natural drug. A computational-based study was carried out to discover and biologically test novel potent and selective HDAC inhibitors (HDACi) from thioester-derivatives and analogues of PsA [46]. Although chemically reduced PsAs and thiol-derived analogues both showed a good inhibitory effect at the nanomolar level in vitro, when they were tested in cancer cell models, their potency was much lower. This biological effect was probably due to the low permeability/stability of thiol in malignant cells. In order to overcome this issue, a novel approach was adopted to “protect” free thiol and enhance its effectiveness in cancer cells. This strategy was developed thanks to the production of novel thioester-active PsA analogues, whose molecular mechanism is mediated by thioester hydrolysis, identified by in vitro assay, and followed by cleavage of the acetyl group by HDAC1 and six enzymes. These newly synthetized thioesters displayed significant cytotoxicity against several cancer cell lines as well as robust enzymatic activity [46]. After confirming the epigenetic role of PsA using in vitro and cell-based assays, a structure–activity relationship (SAR) study was performed by modifying the original scaffold of PsA based on the β-indole-α-oximinoamido protection group and by the replacement of the o-bromophenol unit by an indole ring. These new derivatives were evaluated by several biological assays, displaying cell cycle arrest and p21 induction in acute myeloid leukemia (AML), breast, and prostate cancer cells, as well as histone H3 and alpha tubulin acetylation, showing multiple epigenetic activities [47]. Novel PsA derivatives were synthetized as bisulfide bromotyrosine products, including psammaplins F, G, and H, while two new bromotyrosine derivatives were characterized as psammaplins B, C, D, bisaprasin, I and J, along with the known PsA. [48,49,50,51]. PsA, psammaplin G, and bisaprasin displayed both HDAC and DNMT inhibitory activities, while all the others substantially exhibited HDAC inhibition in vitro.

Given the growing interest in new epigenetic modulators in cancer, research has been focusing on the role of PsA and its derivatives as potential epigenetic markers and investigating their biological activity. Many other computational and biological-based assays have been carried out to optimize the selectivity of psammaplin compounds and determine the best trade-off between chemical stability and epigenetic-based biological function.

#### 4.2.1. UVI5008

The molecular characterization and anticancer activities of UVI5008, a novel synthetic derivative of PsA that exerts multiple epigenetic effects in several cancer cell lines via simultaneous targeting of HDACs, DNMTs, and sirtuins (class III HDACs). UVI5008 is a powerful HDACi, displaying histone H3 acetylation and HDAC inhibition. UVI5008 also inhibits DNA methylation in the promoter region of tumor suppressor gene *pl6INK4a* and alters the acetylation status of chromatin on tumor necrosis factor-related apoptosis-inducing ligand (TRAIL). The inhibitory activity of UVI5008 was also tested on sirtuin 1 and 2, and was found to impact p53 acetylation levels. UVI5008 affects death and ROS pathways in HDAC-resistant and -mutated cancer cells and tumors, providing a potentially valid alternative to combination cancer therapy (patent WO2008125988A1) [52].

#### 4.2.2. Panobinostat

Cinnamic acids play a crucial role in the formation of other more complex phenolic compounds. Panobinostat, (LBH-589; Farydak^®^, Novartis Pharmaceuticals Corporation, East Hanover, NJ, USA), a synthetic analogue of PsA, is one of the most potent pan HDACi and in 2015 received FDA approval for therapeutic application in patients with multiple myeloma (MM). To date, panobinostat has been investigated in numerous completed clinical trials for the treatment of solid and hematological cancers, alone or in combination (NCT01242774, NCT01802879, NCT01336842, NCT01460940, NCT01065467) and about ten clinical studies are currently recruiting (NCT04326764, NCT04341311, NCT02717455, NCT04150289, NCT02386800, NCT02506959, NCT02890069, NCT04315064, NCT01543763, NCT04264143, NCT03143036, NCT03878524). Many reports also confirm the antiangiogenic role of this natural molecule in hepatocellular carcinoma both in vitro and in vivo through the epigenetically regulated connective tissue growth factor [53]. In vitro findings demonstrated that panobinostat inhibits tumor growth in an orthotopic xenograft model of ovarian cancer and that its effect is characterized by acetylation of histone H2B and upregulation of pH2AX, suggesting that these mechanisms are mediated by HDAC inhibition [54]. Panobinostat treatment also showed effective HDAC inhibition in breast, prostate, colon, and pancreatic cancer cell lines, while its effects on normal cells were marginal [55,56], suggesting its cancer-specific selectivity.

#### 4.2.3. NVP-LAQ824

NVP-LAQ824 (dacinostat), another PsA analogue, is a derivative of 4-aminomethylcinnamic hydroxamic acid, which has entered phase I clinical trials [57,58] for the treatment of solid tumors and leukemia. NVP-LAQ824 inhibits HDAC activities and exerts anticancer effects at nanomolar concentrations through a mechanism of action involving the disruption of the charge-relay network via zinc chelation. Dacinostat may interfere with epidermal growth factor-mediated signaling in breast cancer via two independent epigenetic mechanisms involving a decrease in human epidermal growth factor receptor 2 (HER2) mRNA levels and by proteasomal degradation via an increase in the chaperone protein Hsp90. This effect is due to a further increase in acetylation levels induced by a dacinostat-mediated inhibitory mechanism [59]. NVP-LAQ824 was also proposed as a novel HDACi due to its ability to activate p21 at promoter and protein expression level, inhibit cyclin-dependent kinase 2 kinase activity, reduce retinoblastoma phosphorylation, and cause cell cycle arrest selectively in different cancer cell lines and in vivo models [60]. NVP-LAQ824 can also epigenetically modulate macrophage immune response through a mechanism involving recruitment of the transcriptional repressors HDAC11 and PU.1 to the *IL-10* gene promoter. This biological effect results in *IL-10* inhibition and improved responsiveness of CD4+ T cells [61].

#### 4.2.4. Trichostatin A

Trichostatin A (TSA) is an hydroxamic acid originally isolated from the bacterium *Streptomyces platensis*, present in soil, which exerts antifungal, antibacterial, and antineoplastic activities as well as a broad spectrum of reversible HDAC inhibitory functions. Clinical trials investigating TSA in cancer are currently recruiting (NCT03838926, NCT03784417). In a recent study, malignant melanoma cells were treated with TSA and subjected to whole-transcriptome profiling. Data analysis showed that TSA was able to drastically change the transcriptome and several up- and downregulated transcripts were identified within BRAF-mutated melanoma cells. Specifically, TSA was able to downregulate MAPK/MEK/BRAF axis without affecting HDAC and BRAF pathways [62]. Sirtuin 6 is a class III HDAC enzyme involved in various epigenetic-like activities such as gene silencing regulation and DNA repair mechanisms, as well as blood glucose level regulation and stress resistance. Dysregulation of sirtuin 6 has a strong impact in various diseases including dysmetabolism, neurodegeneration, diabetes, and cancer. Since the tumor suppressor protein p53 upregulates sirtuin 6 via a deacetylation mechanism, histone H3 and p53 acetylation (via suppression of sirtuin 6) by TSA has a robust action on cancer cells as the posttranslational modification mediated by acetylation restores the regulation of p53 to normal physiological conditions [63]. TSA analogues such as trichostatic acid, JBIR-109, JBIR-110, and JBIR-111 derive from cultures of the marine sponge-derived *Streptomyces* sp. strain RM72 [64]. The JBIR-17 analogue was instead isolated from *Streptomyces* sp. 26634, in turn isolated from a leaf of the *Kerria japonica* shrub collected in Iwata, Japan [65]. These derivatives display similar biological effects and may be used as lead compounds for the generation of more active drugs.

#### 4.2.5. Vorinostat

Vorinostat (SAHA, ZOLINZA^®^, Merck & Co., Inc., Kenilworth, NJ, USA) was the first HDACi approved by the FDA for the treatment of cutaneous T-cell lymphoma (CTCL) in 2006. Vorinostat is a synthetic derivative of the first natural hydroxamate HDACi identified, TSA, described in the previous subsection [66]. Vorinostat is a pan HDAC inhibitor and the most studied synthetic derivative compound from a natural source. Currently, about 40 clinical trials are in the recruitment phase (the most recent are NCT04308330, NCT04339751, NCT03803605, NCT03056495, NCT02638090, NCT04357873, NCT03167437, NCT03843528, NCT03842696) for a wide variety of diseases. Furthermore, a broad spectrum of datasets present in literature describe and demonstrate the multiple epigenetic roles of vorinostat in cancer and other disorders, as most recently reported in [67,68,69,70,71,72,73,74].

## 5. Cyclic Peptides

Marine-derived secondary metabolites are a huge source of multi-structured peptides possessing unique features able to regulate epigenetic mechanisms in cancer. The majority of these compounds are of natural origin and their backbone, characterized by a ring structure, has been used for the novel synthesis of more active and specific therapeutic drugs. This section mainly discusses depsipeptides and cyclic tetrapeptides (CTPs) as epigenetic-like anticancer agents. Depsipeptides are non-ribosomal peptides in which one or more amine bonds are replaced by the corresponding ester. These derivatives often contain non-protein amino acids and are found in the marine environment. Their synthesis is very straightforward and may lead to the development of several structural combinations useful for identifying the most effective anticancer agents. For instance, modifying amine groups to esters leads to an increase in lipophilicity, thus increasing their cell permeability [75]. Unlike depsipeptides, CTPs are very difficult to synthetize due to their highly complex structure characterized by four amino acids linked by eupeptide bonds. Specifically, CPTs contain l-, d-, and cyclic amino acids, which reduce the cyclic tension associated with CTPs. Many biochemical approaches coupled with extensive studies of three-dimensional structures by X-ray crystallography and NMR have been developed [76] to produce novel and more bioactive molecular structures. The following subsections describe the epigenetic role exhibited by marine-derived cyclic peptides displaying strong anticancer and anticancer-associated biological activities, which may have important implications for human health (Table 2 and Appendix A).

### 5.1. Romidepsin 

Romidepsin ((Istodax^®^, Celgene Corp, Summit, NJ, USA) is a depsipeptide derived from the marine bacterium *Chromobacterium violaceum* and was the first epigenetic-like peptide approved by the FDA for the treatment of CTCL and other peripheral T-cell lymphomas in 2009 and 2011, respectively [77]. Romidepsin is mainly active against class I HDACs via a mechanism involving the release of a thiol by the disulfide bond of the peptide. The resulting mercaptan interacts with zinc at the HDAC binding site, thus inhibiting its activity. Romidepsin is currently the subject of about ten recruiting studies on cancer NCT02512497, NCT01947140, NCT02232516, NCT02616965, NCT03742921, NCT03161223, NCT02783625, NCT04257448, NCT03703375, NCT03593018, NCT02551718). About fifty trials investigating the role of romidepsin in cancer have been completed (the most recent are NCT02296398, NCT01913119, NCT01537744, NCT01324310, NCT01822886, NCT01353664). Other depsipeptide molecules include spiruchostatins [78,79], burkholdacs [80,81], and thailandepsin B [82,83], which are the product of the bacterium *Burkholderia thailandensis*, while FR901375 [84] and largazole [85] are derived from *Pseudomonas chlororaphis* and the cyanobacterium *Symploca* sp., respectively.

### 5.2. Plitidepsin 

Also known as dehydrodidemnin B, plitidepsin (Aplidin^®^, PharmaMar, S.A., Colmenar Viejo, Spain) belongs to the class of didemnins isolated from the tunicate *Aplidium albicans* of the genus *Trididemnum* and is a natural HDACi with a broad spectrum of anticancer effects [86,87]. To date, six studies investigating the anticancer effects of plitidepsin have been completed (NCT01102426, NCT00884286, NCT01149681, NCT02100657, NCT00788099, NCT00229203), five have been terminated (NCT03117361, NCT00780143, NCT03070964, NCT00780975, NCT01876043), and only one is active for patients with COVID-19 (NCT04382066). Plitidepsin displays a strong inhibitory effect on cell growth and apoptosis in MM patients and cell lines, including those resistant to conventional therapies. This bioproduct also potently inhibits osteoclast differentiation and bone resorptive activity both in vivo and in vitro [88]. Among hundreds of plitidepsin analogues, PM01215 and PM02781 (patent WO 2002002596) were identified for their antiangiogenic effect in human primary cells [89].

### 5.3. Largazole

Largazole is a macrocyclic depsipeptide deriving from the marine cyanobacterium *Symploca* sp. [90]. This molecule is considered a superior hybrid thanks to its structural characteristics: it contains a thiazole unit linked to a 4-methylthiazoline, a nonmodified L-valine amino acid, and a thioester responsible for its mechanism of action. Largazole acts as an HDACi and is particularly active in colon cancer cell lines, as documented by a screening of 60 cell lines from the National Cancer Institute. In vivo and in vitro studies showed its apoptotic and antiproliferative activities as well as histone H3 hyperacetylation. Many similarities in terms of gene regulation were also found with two other potent HDACi, vorinostat and FK228, by gene transcriptomic profiling [85,91].

### 5.4. Azumamides

Azumamides are a group of CTPs isolated from the Japanese marine sponge *Mycale izuensis*, with five isoforms (A, B, C, D, E). Azumamides A–E were the first cyclic peptides with HDAC inhibitory activity isolated from marine organisms, and are characterized by four non-ribosomal amino acid residues, three of which are D-series amino acids while only one is a beta amino acid [92]. Azumamides A–E were identified for the first time as potent HDACi in a chronic myeloid leukemia cell line [93] following in vitro evaluation of HDAC activity. Specifically, increasing concentrations of azumamide A induced histone H3 acetylation while producing cytotoxic effects in colon cancer and chronic leukemia cells. Azumamide variants B, C, and E produced an HDAC inhibitory effect in human carcinoma cell lines with IC50 values in the micromolar range. Derivative E resulted the most active compound with inhibitory activities due to its different chemical structure represented by a carboxylic acid, which has a higher affinity for thee HDAC active site containing zinc ion, unlike the amide group present in the other azumamides A, B, and D [94]. From a mechanistic perspective, azumamide E was the only isoform found able to induce overexpression of p21, a well-known marker regulating cell cycle progression, in murine induced pluripotent stem cells [95].

### 5.5. Trapoxins

Trapoxin (TPX) A is a fungal-derived HDACi with a homodetic cyclic tetrapeptidic structure isolated from the species *Helicoma ambiens*. This molecule is an epoxyketone and exerts irreversible inhibitory effects on class I HDACs due to the analogous structure of its ketone carbonyl group and the carbonyl of the substrate acetyl-L-lysine of HDACs. A study reporting the creation of a novel X-ray structure characterized by trapoxin A bound to HDAC8 demonstrated that trapoxin A is a non-covalent HDAC8 inhibitor thanks to an α,β-epoxyketone side chain, which by chemical transition state is able to bind the HDAC active site containing zinc [96]. Cyclic hydroxamic acid-containing peptide (CHAP) 1 is a hybrid compound deriving from trapoxin A and TSA, in which the epoxyketone group is substituted by the hydroxamic acid instead of the epoxyketone and can reversibly inhibit HDACs at low nanomolar concentrations. Although several CHAP derivatives have been produced, only one showed antitumor activity in BDF1 mice bearing B16/BL6 tumor cells, suggesting the possibility of an improved synthesis of new hybrids [97].

### 5.6. Apicidin

Apicidin is a fungal metabolite derived from the species *Fusarium pallidoroseum*, in the *Sordariomycetes* class. It is an HDAC2 and 3 inhibitor and acts as a trapoxin A analogue, but lacks the epoxyketone functional group. Several studies described the anticancer activities of apicidin in vitro and in vivo [42,98,99]. A recent report investigated the characterization of HDAC3 in Notch signaling by comparing data obtained following apicidin treatment and HDAC3 loss of function in APRE T cell models. Gene expression data from RNA sequencing revealed a cluster of 65 upregulated genes and another of 368 downregulated genes in both HDAC knockdown and apicidin-treated cells. Many of the identified downregulated genes affected Notch signaling, and in particular apicidin treatment led to an increase in epigenetic markers such as acetylation levels of histone H3 lysine 27 (H3K27), histone H3 lysine 18 (H3K18), and histone H3 lysine 9 (H3K9), while a decrease in H3K27 acetylation was detected at the recombination signal binding protein for immunoglobulin kappa J region binding sites associated with Notch target genes. Apicidin treatment affected NOTCH1 intracellular domain stability via a mechanism driven by proteasomal degradation mediated by ubiquitination [100]. Microsporins A and B are also marine-derived metabolites from the fungus *Microsporum* cf. *gypseum* that display HDACi action. Their cytotoxic-related activities were reported in colon adenocarcinoma cells and a precursor to the unusual amino acid residue of the anticancer agent microsporin B was subsequently synthetized as (*S*)-2-Boc-Amino-8-(*R*)-(tert-butyldimethylsilanyloxy) decanoic acid [101].

## 6. Alkaloids

Alkaloids are natural compounds characterized by a nitrogen-heterocyclic structure. Specifically, marine alkaloids have an amine nitrogen group and a carbon ring and mainly derive from marine organisms such as sponges, algae (green, brown, and red), coelenterates, and tunicates. These metabolites display several properties, acting as antitumor, antiviral, antimalarial, antifungal, and anti-osteoporosis agents. Marine alkaloids may be used as chemotherapeutics or as lead compounds for structural modification (Table 2).

### 6.1. Brominated Alkaloids: Isofistularin-3

Brominated alkaloids (BAs) include the promising natural molecule isofistularin-3 (Iso-3), whose source is the sponge Aplysina aerophoba. Structurally, this compound shows similarities with PsA, a well-known bromotyrosine derivative. Iso-3 was screened for its DNMT1 inhibitory activities in vitro together with a library of compounds. The conformational structure of this compound was also analyzed by molecular docking prediction, revealing an inhibition interaction between DNMT1 and DNA via a conserved CXXC motif affecting binding activity via positively charged residues. BAs lack a thiol linker moiety, explaining the absence of HDAC inhibitory activity. Iso-3 was shown to have anticancer potential in lymphoma cells, leading to cell cycle arrest, morphological changes, and authophagy as well as caspase-dependent and -independent cell death [33].

### 6.2. Bispyridinium Alkaloids: Cyclostellettamines

Marine-derived alkaloids include a group of compounds with a macrocyclic ring of the precursor bispyridinium alkaloid called cyclostellettamines. Cyclostellettamines A and G together with dehydrocyclostellettamine D and E, shown to act as HDAC inhibitors in the myelogenous leukemia K562 cell line, were isolated from a marine sponge of the genus Xestospongia [102]. The inhibitory effect of these compounds was very weak, with cytotoxic activities observed in human cervix carcinoma, mouse leukemia, and rat fibroblasts, suggesting that multifunctional targets of these molecules can modulate their cytotoxic effects. A synthetic route for cyclostellectamines A–L and dehdrocyclostellettamines D and E was developed using bispyridinium dienes precursors and subsequent catalytic hydrogenation. The compounds obtained and their precursors were tested in vitro in an AML cell line for HDAC activity, cell cycle modulation, acetylation levels of histone H3 and tubulin, differentiation, and apoptosis. The precursors were found to have more potent activities than the natural compounds [103].

## 7. Conclusions

Anticancer therapy-associated drawbacks include resistance to drug treatments and the occurrence of relapses, whereby, finding and characterizing new drugs is one of the main objectives. The development of new drugs with anticancer activity follows a multidisciplinary approach that generally begins with the identification and retrieval of new bioactive molecules from natural sources, which in turn undergo preliminary evaluations assessing biological activity, toxicological tests, and chemical/biotechnological synthesis [23]. The difficulty in finding natural substances of marine origin and collecting sufficient quantities for clinical and preclinical experimentation often hinders the possibility of isolating natural biomolecules, preventing the development of promising compounds. Although the epigenetic role of natural compounds has been discussed in previous studies, the aspects related to the discovery of new marine-derived anticancer bio-compounds highlighting the variability that characterizes the organisms themselves and their surrounding environment that, have not been extensively discussed previously. The chemical-physical-biological characteristics of natural marine compounds are unique and cannot be found in the terrestrial environment, but the properties of already characterized molecules can be exploited for a new chemical synthesis and molecular modeling of new products to refine their anticancer activity. Natural marine bio-compounds are produced in co-evolution with biological systems and can be specific mediators of epigenetic processes in cancer, in turn influencing the abundance and distribution of species in nature and the functioning of ecosystems. A very important aspect also involves the concept of environmental sustainability, linked to the strong need to reduce the impact of the ecosystem on natural resources and the need to safeguard the marine environment to maintain and preserve biodiversity and prevent as much as possible the decrease of ecosystem functions. Marine biotechnologies are increasingly specializing in the development of new methods based on the evaluation of the sustainability of organisms sampling for their subsequent use associated with new selection criteria and the creation of marine biobanks.

Marine organisms produce secondary metabolites whose chemical-physical and biological characteristics are extremely variable due to biotic and abiotic factors, adding a further level of complexity to research and development efforts in this field. Most compounds of marine origin can be synthesized and more than 10 are currently at an advanced clinical stage [104]. Furthermore, new and advanced technologies allow the biotechnological production of these molecules either through cloning techniques or gene cluster manipulation, overcoming a number of obstacles including those linked to environmental risks associated with the potential loss of genetic resources caused by overharvesting of producer organisms. Major interest is currently focusing on the identification and biosynthetic characterization of natural marine compounds, particularly those derived from the secondary metabolism, potentially available as active principles (lead compounds) or biochemically comparable (biosynthetic analogues) to active compounds, for the development of new epigenetic drugs. Many marine compounds with anticancer activity capable of modulating microRNA and epigenetic mechanisms such as DNA methylation, acetylation, and histone methylation, have a considerable impact on the regulation of gene expression [105]. Marine organisms are themselves subject to intrinsic epigenetic changes induced by the surrounding environment, causing them to produce biomolecules with unique structural characteristics that can act as an imprint to produce a novel synthesis. An example of a response to ecological changes is represented by dimethylsulfoniopropionate (DMSP), a metabolite that can be degraded by phytoplankton or bacteria to produce dimethylsulfide (DMS). Inducing the bloom of the Gulf of Mexico phytoplankton, bacterioplankton cells can demethylate this metabolite via the dmdA gene pointing out several dmdA subclases identified in response to ecological alteration [106]. Epigenetic mechanisms such as DNA methylation and histone modifications may also affect coral adaptation to climate change [107] spreading to subsequent generations, but these mechanisms still need to be further studied [108]. The chromosomal characterization of different species of sponges has been carried out on the basis of their phylogenetic relationships, identifying similar karyotypes, harboring a diploid chromosome number. A high variability in the extent of the genome has been defined also in species belonging to the same class, reflecting distinct genomic organization [109]. Further studies will allow to understand how epigenetic mechanisms, which are sometimes stochastic events and often are responsible for locus-specific gene expression via chromatin modifications, can be correlated with organism ploidy. As cancer treatments are increasingly based on personalized medicine due to the complexity of the disease and the multitude of hallmarks involved, including epigenetic alterations, developing new bioactive molecules derived from marine sources will provide a vast repertoire of substances with pharmacological activity that can be used alone or in combination with other epigenetic drugs, chemotherapy, or radiotherapy.

## Figures and Tables

**Figure 1 marinedrugs-19-00015-f001:**
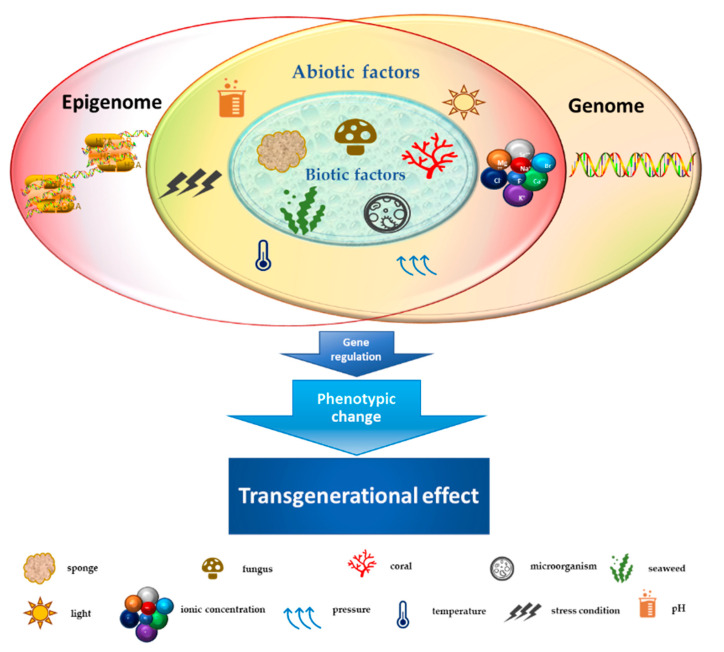
Schematic representation of the crosstalk between biotic/abiotic factors and transgenerational genetic/epigenetic effects.

**Figure 2 marinedrugs-19-00015-f002:**
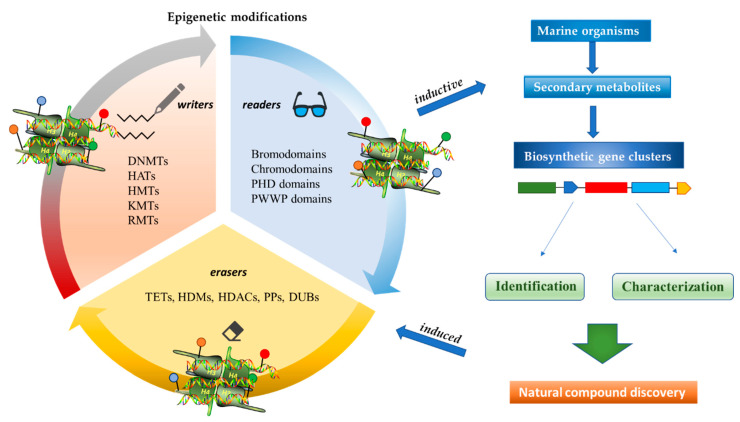
Inductive and induced epigenetic modifications by secondary metabolites produced by marine organisms. Epigenetic writers, readers, and erasers regulate production of secondary metabolites, which in turn induce epi-modifications.

**Figure 3 marinedrugs-19-00015-f003:**
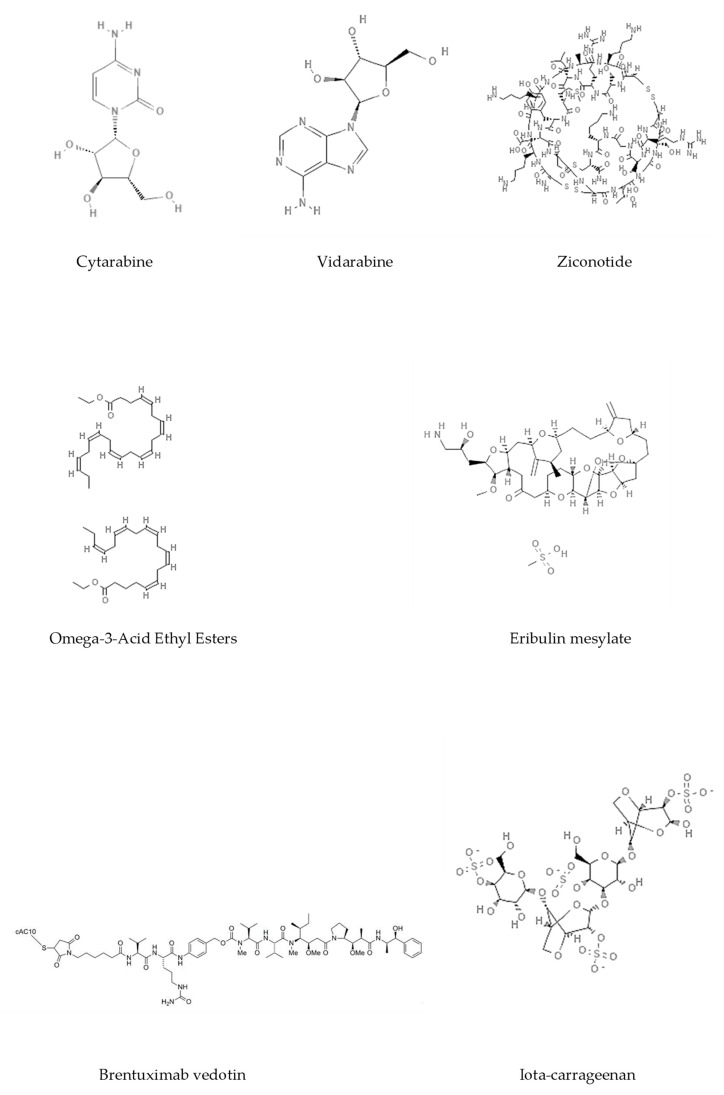
Chemical structures of marine-derived therapeutic compounds approved by the U.S. Food and Drug Administration (FDA).

**Table 1 marinedrugs-19-00015-t001:** Natural phenolic compounds and their derivatives, and their epigenetic role in cancer.

Compound	Structural Formula	Chemical Class	Source	Species	Epigenetic Mechanism	Ref
**Psammaplin A**	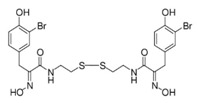	Phenolic compound	Sponge	*Psammaplin aplysilla*, *Thorectopsammaxana*	DNMT inhibition (in vitro)	[36,44,47,50]
					HDAC inhibition (in vitro)	[37,42,43,46,47,57,58,59,61,64,66]
					Topoisomerase II inhibition (in vitro)	[39]
					Inhibition of DNA regulation (in vitro)	[36]
					Aminopeptidase N inhibition (in vitro)	[40]
					SIRT1 induction (in vitro)	[47]
					HDAC inhibition (in vitro)	[46]
					Increased H3 acetylation (in vitro)	
**Psammaplin F**	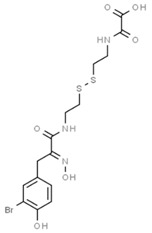	Phenolic compound	Sponge	*Pseudoceratina purpurea*	HDAC inhibition (in vitro)	[50]
**Psammaplin G**	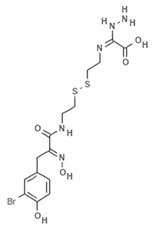	Phenolic compound	Sponge	*Pseudoceratina purpurea*	DNMT inhibition (in vitro)	[50]
**Bisaprasin**	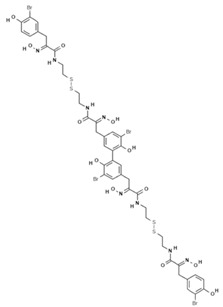	Phenolic compound	Sponge	*Pseudoceratina purpurea*	DNMT inhibition (in vitro)	[50]
**UVI5008**	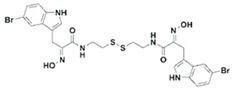	Phenolic compound		*Psammaplin derivative*	DNMT3a inhibition (in vitro)	[52]
					H3 hyperacetylation (ex vivo)	
					HDAC inhibition (in vitro)	
					HDAC1–4 inhibition (in vitro)	
					SIRT inhibition (in vitro)	
**NVP-LAQ824 (dacinostat)**	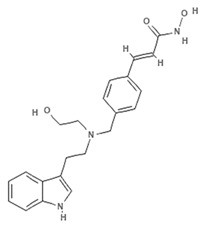	Hydroxamic acid		*Psammaplin derivative*	HDAC inhibition (in vitro)	[57,59,60]
					HDAC inhibition (in vivo)	[57,58]
					Increased H3 acetylation (in vitro)	[59,60,61]
					Increased H4 acetylation (in vitro)	[60,61]
**Trichostatin A**	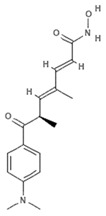	Hydroxamic acid	Bacterium	*Streptomyces platensis*	HDAC inhibition (in vitro)	[62,66]
					MAPK/MEK/BRAF downregulation (in vitro)	[62]
					Increased H3 acetylation (in vitro)	[63]
**JBIR-109**	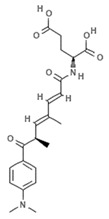	Trichostatin analogue	Sponge	*Streptomyces* sp. strain RM72	HDAC inhibition (in vitro)	[64]
**JBIR-110**	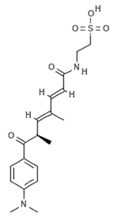	Trichostatin analogue	Sponge	*Streptomyces* sp. strain RM72	HDAC inhibition (in vitro)	[64]
**JBIR-111**	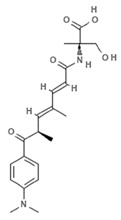	Trichostatin analogue	Sponge	*Streptomyces* sp. strain RM72	HDAC inhibition (in vitro)	[64]
**JBIR-17**	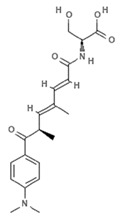	Phenolic compound	Bacterium	*Kerria japonica*	HDAC inhibition (in vitro)	[65]
**Panobinostat**	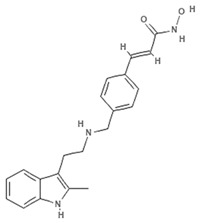	Phenolic compound	Sponge	*Psammaplin aplysilla*	Pan-HDAC inhibition (in vitro)	[53,54]
					HDAC inhibition (in vitro)	[55]
**Vorinostat**	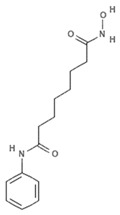	Hydroxamic acid		*Trichostatin A derivative*	Pan HDAC inhibition (in vitro)	[67]
					HDAC inhibition (in vitro)	[66,69,70,73]
					HDAC inhibition (in vivo)	[67,68]
					mTOR inhibition (in vivo)	[68]
					PLD-1 upregulation (in vitro)	[70]

**Table 2 marinedrugs-19-00015-t002:** Cyclic peptides and alkaloids with their epigenetic anticancer role.

Compound	Structural Formula	Chemical Class	Source	Species	Epigenetic Mechanism	Ref
**Romidepsin**	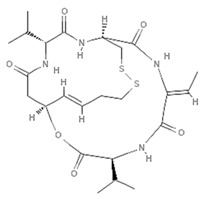	Depsipeptide	Bacterium	*Chromobacterium violaceum*	HDAC inhibition (in vitro)	[80,82,83,84]
**Plitidepsin**	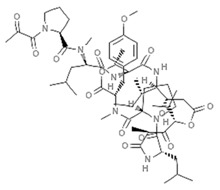	Cyclic tetrapeptide	Tunicate	*Aplidium albicans*	Caspase-3 upregulation (in vitro)	[87]
					Dephosphorylation of ERK1/2 and 5 (in vitro)	[88]
**PM01215**	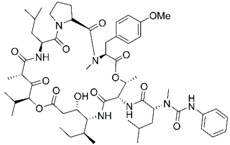		Aplidin analogues		p16INK4A induction (in vitro)	[89]
					VEGF downregulation (in vitro)	
**PM02781**	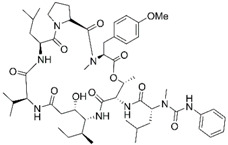		Aplidin analogues		p16INK4A induction (in vitro)	[89]
**Largazole**	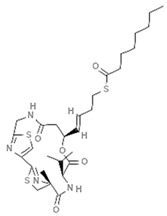	Macrocyclic depsipeptide	Cyanobacterium	*Symplocasp*	HDAC inhibition (in vitro)	[85,91]
					HDAC inhibition (in vivo)	[85]
**Azumamides**	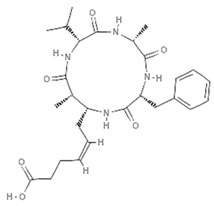	Cyclic tetrapeptide	Sponge	*Mycale izuensis*	HDAC inhibition (in vitro)	[92,93,95]
					Increased H3 acetylation (in vitro)	[94]
**Trapoxins**	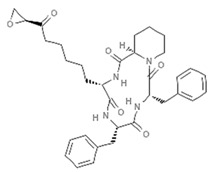	Cyclic tetrapeptide	Fungus	*Helicoma ambiens*	Class I HDAC inhibition (in vitro)	[96]
					HDAC inhibition (in vitro)	[97]
**Apicidin**	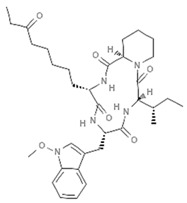	Cyclic tetrapeptide	Fungus	*Fusarium pallidoroseum*	HDAC8 inhibition (in vivo)	[98]
					p21 upregulation (in vitro)	[99]
					HDAC inhibition (in vitro)	[99,100]
					DNMT inhibition (in vitro)	[42]
					HDACs2/3 inhibition (in vitro)	[42]
**Microsporins A and B**	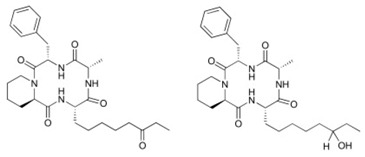	Cyclic peptide	Fungus	*Microsporum* cf. *gypseum*	HDAC inhibition (in vitro)	[101]
					HDAC8 inhibition (in vitro)	
**Isofistularin-3**	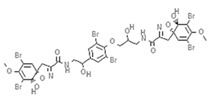	Alkaloid	Sponge	*Aplysina aerophoba*	DNMT1 inhibition (in vitro)	[33]
**Cyclostellettamines**	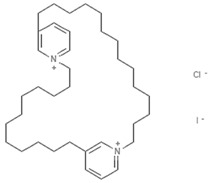	Alkaloid	Sponge	*Xestospongia*	HDAC inhibition (in vitro)	[102,103]

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
