# Peer review of "Marine-Derived Secondary Metabolites as Promising Epigenetic Bio-Compounds for Anticancer Therapy"

_marinedrugs, 2020, doi:10.3390/md19010015_

Round 1

Reviewer 1 Report

The authors reviewed marine-derived secondary metabolites for anticancer therapy. However, this manuscript in not recommended for publication in its present form, but may be reconsidered as a new paper after throughout revisions.

  1. The authors should mention whether a review article similar to the content of this manuscript has been published in the past.
  2. The authors should include why they have put together a review on this topic in the manuscript.
  3. In this manuscript, the authors do not explicitly state the scope of this review. Dose the content of this manuscript treat research results in this area comprehensively? It should also be mentioned the relevance of the content of this manuscript to the authors' work.
  4. This manuscript is inconvenient because it does not have a structural formula for the compound. The authors should add the structural formulas of all the compounds covered in this manuscript.

Author Response

We are grateful to the reviewer for the insightful comments.

We modified the text to fulfil the suggestions provided by the reviewer. Changes have been highlighted within the manuscript.

Here follows a point-by-point response to the reviewer comments and concerns

Response to Reviewer 1 Comments

The authors reviewed marine-derived secondary metabolites for anticancer therapy. However, this manuscript in not recommended for publication in its present form, but may be reconsidered as a new paper after throughout revisions.

  1. The authors should mention whether a review article similar to the content of this manuscript has been published in the past.

We thank the reviewer very much for this very important suggestion. We addressed this point in the Conclusion section (lanes 456-459)

  1. The authors should include why they have put together a review on this topic in the manuscript.

Thank you very much for your suggestion. You have raised an important point here. We addressed this point in Conclusion section (lanes 448-449 and lanes 459-471)

  1. In this manuscript, the authors do not explicitly state the scope of this review. Dose the content of this manuscript treat research results in this area comprehensively? It should also be mentioned the relevance of the content of this manuscript to the authors' work.

We agree with this comment and have incorporated your suggestion in the manuscript. We have added a new sentence in lanes 61-64 and we have also added a new section in the manuscript: Sustainability and Health” to strengthen the aim of this review. The new section has been renamed as section 3 within the manuscript (lanes 155-168)

  1. This manuscript is inconvenient because it does not have a structural formula for the compound. The authors should add the structural formulas of all the compounds covered in this manuscript.

We apologize for missing this important point. A new figure has been inserted into the manuscript with the structural formulas of FDA-approved, marine-derived therapeutic compounds. The new figure has been renamed as Figure 3, following the manuscript. Structural formulas of all other compounds covered in the manuscript, have been added in Tables 1 and Tables 2.

Reviewer 2 Report

Referee report to the manuscript entitled:

Marine-derived secondary metabolites as promising 2 epigenetic bio-compounds for anticancer therapy

By Mariarosaria Conte, Elisabetta Fontana, Angela Nebbioso and Lucia Altucci

The manuscript tackles a very interesting and only inadequately discussed issue, the effect of marine secondary metabolites on the epigenetic memory. This process is defined – also along the authors – as reversible heritable changes in gene expression that are not due to alterations in DNA sequence. This implies that methylation and demethylation processes and histone modifications are main tasks to be performed. This is what the authors did perfectly.

The questions which arise always with the attribution of process to an epigenetic level is, that no correlations to the chromatin structure and the DNA modifications have been performed. I know this very –very difficult to assess in those lower organisms.

+) A solution might be possible if perhaps genes from seaweeds, corals, and/or sponges have been isolated – I would guess no. So, a solution would be to check in the literature if these marine organisms have the responsible enzymes as activities. Here the authors could be successful to find. As an example: Howard, E. C., Sun, S., Reisch, C. R., Del Valle, D. A., Bürgmann, H., Kiene, R. P., et al. (2011). Changes in dimethylsulfoniopropionate demethylase gene assemblages in response to an induced phytoplankton bloom. Appl. Environ. Microbiol. 77, 524–531. doi: 10.1128/AEM.01457-10. Perhaps the authors can comment on those findings and if nothing else is available they can discuss this in the text.

+) The authors can also check the paper of: Bourne DG, Morrow KM, Webster NS. Insights into the Coral Microbiome: Underpinning the Health and Resilience of Reef Ecosystems. Annu Rev Microbiol. 2016 Sep 8;70:317-40. doi: 10.1146/annurev-micro-102215-095440. Epub 2016 Jul 8. PMID: 27482741. Then they might get an idea how histone modification enzymes could be involved.

+) After that, the author can draw a sketch and hypothesize which factors influence the gene expression systems in the sponges et al – epigenetically.

+) Is anything known about the histone code in those lower animals??? See Quina AS, Buschbeck M, Di Croce L. Chromatin structure and epigenetics. Biochem Pharmacol. 2006 Nov 30;72(11):1563-9. doi: 10.1016/j.bcp.2006.06.016. Epub 2006 Jul 11. PMID: 16836980.

+) In the meantime the chromatin and gene structure of sponges are known. See also: Ishijima J, Iwabe N, Masuda Y, Watanabe Y, Matsuda Y. Sponge cytogenetics - mitotic chromosomes of ten species of freshwater sponge. Zoolog Sci. 2008 May;25(5):480-6. doi: 10.2108/zsj.25.480. PMID: 18558800. Therefore, it could be discussed if polyploidy is correlated with the epigenetic “program”.

The paper – as it stands – is certainly of very high level. However, I would be grateful if the expert authors could comment also on the gene expression system in their epigenetic approach.

Author Response

We are grateful to the reviewer for the insightful comments.

We modified the text to fulfil the suggestions provided by the reviewer. Changes have been highlighted within the manuscript.

Here follows a point-by-point response to the reviewer comments and concerns

Response to Reviewer 2 Comments

The manuscript tackles a very interesting and only inadequately discussed issue, the effect of marine secondary metabolites on the epigenetic memory. This process is defined – also along the authors – as reversible heritable changes in gene expression that are not due to alterations in DNA sequence. This implies that methylation and demethylation processes and histone modifications are main tasks to be performed. This is what the authors did perfectly.

 The questions which arise always with the attribution of process to an epigenetic level is, that no correlations to the chromatin structure and the DNA modifications have been performed. I know this very –very difficult to assess in those lower organisms.

+) A solution might be possible if perhaps genes from seaweeds, corals, and/or sponges have been isolated – I would guess no. So, a solution would be to check in the literature if these marine organisms have the responsible enzymes as activities. Here the authors could be successful to find. As an example: Howard, E. C., Sun, S., Reisch, C. R., Del Valle, D. A., Bürgmann, H., Kiene, R. P., et al. (2011). Changes in dimethylsulfoniopropionate demethylase gene assemblages in response to an induced phytoplankton bloom. Appl. Environ. Microbiol. 77, 524–531. doi: 10.1128/AEM.01457-10. Perhaps the authors can comment on those findings and if nothing else is available they can discuss this in the text.

Thank you very much for your kind suggestions. We have discussed this important point within the text (lanes 488-492.)

+) The authors can also check the paper of: Bourne DG, Morrow KM, Webster NS. Insights into the Coral Microbiome: Underpinning the Health and Resilience of Reef Ecosystems. Annu Rev Microbiol. 2016 Sep 8;70:317-40. doi: 10.1146/annurev-micro-102215-095440. Epub 2016 Jul 8. PMID: 27482741. Then they might get an idea how histone modification enzymes could be involved.

+) After that, the author can draw a sketch and hypothesize which factors influence the gene expression systems in the sponges et al – epigenetically.

We agree with this point and have incorporated your suggestion throughout the manuscript (lanes 492-493-525)

+) Is anything known about the histone code in those lower animals??? See Quina AS, Buschbeck M, Di Croce L. Chromatin structure and epigenetics. Biochem Pharmacol. 2006 Nov 30;72(11):1563-9. doi: 10.1016/j.bcp.2006.06.016. Epub 2006 Jul 11. PMID: 16836980.

Thank you for pointing this out. We have added the relative reference to strengthen this point in (see lanes 494)

+) In the meantime the chromatin and gene structure of sponges are known. See also: Ishijima J, Iwabe N, Masuda Y, Watanabe Y, Matsuda Y. Sponge cytogenetics - mitotic chromosomes of ten species of freshwater sponge. Zoolog Sci. 2008 May;25(5):480-6. doi: 10.2108/zsj.25.480. PMID: 18558800. Therefore, it could be discussed if polyploidy is correlated with the epigenetic “program”.

 We thank the reviewer very much for this very important suggestion. We have added this point in lanes :494-501

Round 2

Reviewer 1 Report

This revised manuscript has been modified according to the reviewer's comments. It is acceptable for publication.